# The Role of Gut Microbiota in the Skeletal Muscle Development and Fat Deposition in Pigs

**DOI:** 10.3390/antibiotics11060793

**Published:** 2022-06-11

**Authors:** Qi Han, Xingguo Huang, Fuyong Yan, Jie Yin, Yingping Xiao

**Affiliations:** 1College of Animal Science and Technology, Hunan Agriculture University, Changsha 410128, China; 18404985053@163.com (Q.H.); hxg68989@hunau.edu.cn (X.H.); yinjie@hunau.edu.cn (J.Y.); 2Hunan Jiuding Technology (Group) Co., Ltd., Changsha 410007, China; 3State Key Laboratory for Managing Biotic and Chemical Threats to the Quality and Safety of Agro-Products, Institute of Quality and Standard for Agro-Products, Zhejiang Academy of Agricultural Sciences, Hangzhou 310021, China

**Keywords:** gut microbiota, pork, muscle fiber, fat deposition, host genetics, diet composition

## Abstract

Pork quality is a factor increasingly considered in consumer preferences for pork. The formation mechanisms determining meat quality are complicated, including endogenous and exogenous factors. Despite a lot of research on meat quality, unexpected variation in meat quality is still a major problem in the meat industry. Currently, gut microbiota and their metabolites have attracted increased attention in the animal breeding industry, and recent research demonstrated their significance in muscle fiber development and fat deposition. The purpose of this paper is to summarize the research on the effects of gut microbiota on pig muscle and fat deposition. The factors affecting gut microbiota composition will also be discussed, including host genetics, dietary composition, antibiotics, prebiotics, and probiotics. We provide an overall understanding of the relationship between gut microbiota and meat quality in pigs, and how manipulation of gut microbiota may contribute to increasing pork quality for human consumption.

## 1. Introduction

Pork, a vital source of animal protein for human growth and development, is one of the most commonly consumed meats in the world. According to the UN’s Food and Agriculture Organization, a powerful growth with an expected increase of 13% by 2030 will occur in pig husbandry [1], indicating that the pig industry plays a crucial role in the food supply chain and possesses a high economic impact. It is well known that pork quality determines the purchase decision of consumers and affects producers and retailers. However, meat quality defects still exist, which bring great economic losses to the fresh meat market [2]. Hence, guaranteeing an adequate supply of safe and high-quality pork has increasingly attracted global attention [3]. Meat quality is a complicated trait, which is controlled by multiple factors and mechanisms. Although there are many research studies on many different aspects of pork quality, unexpected variation in pork quality remains a major problem in the meat industry. The key parameters of pork quality, including color, pH, water holding capacity, tenderness, flavor, and juiciness, are mainly associated with the development of muscle fiber, the type of muscle fiber and the deposition of intramuscular fat. Thus, more insight into the factors and related mechanisms affecting pigs muscle development and fat deposition are needed to improve meat quality.

Like other farm animals and humans, the gastrointestinal tract of pigs contains trillions of commensal microorganisms, mainly bacteria, which constitute the gut microbiota and are closely associated with host physiological functions [4], feed efficiency, lipid metabolism, immune system development, and gut health [5,6,7,8]. Recently, gut microbiota have been considered as an important factor affecting complicated traits such as meat quality. Indeed, accumulating evidence suggests that gut microbiota regulate muscle development and fat deposition through metabolite–host interactions in humans [9,10], mice [11,12], and farm animals [4,6,13]. Notably, a variety of environmental factors, such as diet [14,15], stress, and medication [16], can govern the composition of the gut microbiota, thus providing a unique opportunity to improve the physiological condition of the host by regulating gut microbial communities, which may improve the production of high-quality meat. This review focuses mainly on how gut microbiota affect muscle growth, development, and fat deposition in pigs, as well as the factors influencing gut microbiota composition and diversity, providing a vital theoretical basis for future research.

## 2. Gut Microbiota Linked to Muscle Growth and Development

Skeletal muscle accounts for approximately 50% of body mass in mammals and its growth and development affect indirectly the quantity and quality of pork (pH, meat color, drip loss, tenderness, and juiciness) [17,18,19]. In general, muscle growth and development are governed by multiple factors, such as breed, genotype, sex, diet, muscle location, hormones, exercise, and ambient temperature. Interestingly, recent studies have suggested that stable and diverse gut microbiota can govern muscle development, growth, and function [20,21,22]. Therefore, it is considered that the gut microbiota are expected to become potential biological targets for increasing muscle quantity and improving muscle quality.

### 2.1. Muscle Mass and Composition

Gut microbiota are involved in maintaining skeletal muscle mass. For example, germ-free mouse skeletal muscle showed a reduction in skeletal muscle mass compared to the pathogen-free mouse skeletal muscle. However, the mouse muscle mass lost could be restored partially following fecal microbiota transplantation from pathogen-free mice into germ-free mice [11,23]. Similarly, germ-free piglets exhibit lower body weight and lean mass than normal piglets with intestinal microbiota [20]. Fiber type composition is markedly different between muscles, both within and between animals. In general, muscle fiber is divided into four types, namely type I or slow-oxidative, type IIA or fast oxidative-glycolytic, and fast glycolytic IIX or IIB, which are encoded by *MYH7*, *MYH2*, *MYH1*, and *MYH4* genes, respectively [24]. Importantly, increasing the proportion of slow-twitch or oxidative muscle fibers with a higher ability of oxidative metabolism contributes to better meat quality than those with fast-twitch or glycolytic fibers [25]. Growing evidence indicates that gut microbiota exert profound effects on muscle fiber formation. The relationship between the gut microbiota and muscle fiber has been unraveled using fecal microbiota transplantation. After receiving a gut microbial community obtained from obese Rongchang pigs, germ-free mice displayed a higher slow-contracting fiber level, a reduced fiber size, and fast IIB fiber ratio [26]. Moreover, Qi et al. [20] found that germ-free piglets colonized with microbiota harvested from normal piglets can be reliably expected to restore the slow-twitch muscle fibers partially in piglets. Such findings underscore a role for gut microbiota in influencing muscle fiber characteristics and open the door to study the effect of gut microbiota on meat quality. The mechanisms underlying the association between the microbiome and skeletal muscle function have been gradually found. More recently, bacterial metabolite contributions to the transformation of muscle fiber types have also been studied [20].

Intramuscular fat (IMF), deposited among muscle fibers or within muscle cells, is a vital factor influencing pork quality. With the development of omics technology, the gut microbiota associated with IMF deposition have been gradually elucidated. For instance, Fang et al. revealed that the vast bulk of the IMF-associated operational taxonomic units (OTUs) belonged to bacterial genera such as *Prevotella*, *Treponema*, *Bacteroides,* and *Clostridium*, which are involved in polysaccharide degradation and amino acid metabolism [27]. A further study revealed that some genera (such as *Prevotellaceae UCG-001*, *Alistipes* in the cecum and *Clostridium sensu stricto 1* in the jejunum) were identified to be highly positively correlated to IMF by 16S rRNA gene sequencing [28]. Further investigation indicated that mice receiving Jinhua pig-derived microbiota enhanced IMF deposition [29]. The above observations suggest that gut microbiota play a pivotal role in porcine intramuscular adipogenesis.

### 2.2. Effects of SCFAs on Skeletal Muscle Metabolism

Short-chain fatty acids (SCFAs) (principally acetate, propionate, and butyrate) are gut-derived metabolites produced by anaerobic microbes through fermentation of indigestible dietary fiber in the host [30]. These SCFAs facilitate glucose uptake and promote glycogen synthesis in skeletal muscle. Studies in vitro showed that acetate or propionate administration can enhance insulin-independent glucose uptake in L6 or C2C12 myotubes, respectively [31,32]. Furthermore, emerging studies of various murine models have suggested that acetate also contributes to glycogen synthesis by inhibiting glycolysis and reducing the consequent accumulation of glucose 6-phosphate [33,34,35]. In addition, acetate treatment increased skeletal muscle Glut4 mRNA and protein levels in L6 myotubes and rats [31,36]. Importantly, Glut4, a primary glucose transporter protein, exerts an essential role in glucose uptake and metabolism into skeletal muscle [37]. Therefore, the glucose metabolism-related effects of SCFAs might be due to the Glut4 expression.

It is known that changes in protein metabolism ultimately result in the changes in skeletal muscle mass and phenotype. Therefore, changes in mass and phenotype can be used to infer changes in protein metabolism within skeletal muscle. Mounting evidence suggested that SCFAs supplementation influences skeletal muscle mass in mice [31,38,39]. Moreover, emerging evidence points out that the addition of SCFAs contributes to the formation of oxidative skeletal muscle phenotype, including enhanced oxidative capacity, increased mitochondrial content and a higher proportion of type I in rodents [38,40]. Similarly, a study on germ-free piglets showed that the decreased SCFAs content in circulation in muscle can, at least in part, result in the reduction in slow-twitch muscle fibers proportion [20], and the potential mechanisms are likely to be through upgrading the expression of myofiber type-related microRNAs and PGC-1α, eventually contributing towards good pork quality [41]. The observations mentioned above highlight the importance of SCFAs in protein metabolism within skeletal muscle. Nevertheless, the underlying mechanisms for this effect requires further exploration.

Similarly, the direct impacts of the SCFAs on skeletal muscle lipid metabolism have been documented in vivo and in vitro studies. In L6 myotubes, butyrate or acetate treatment was able to enhance fatty acid oxidation or uptake, respectively [31,42]. Studies performed on mice have demonstrated that the addition of butyrate also increased fatty acid oxidation and decreased the concentration of triglyceride and cholesterol within skeletal muscle [39,42,43]. Likewise, a pig-to-mouse fecal microbiota transplantation (FMT) study demonstrated that FMT from obese Jinhua pigs into recipient mice caused an accumulation of lipid and triglyceride, and increased the lipoprotein lipase activity in gastrocnemius muscles, which can be attributed, in part, to the reduction in acetate and butyrate of colon. Accordingly, SCFAs are of great importance in modulating skeletal muscle lipid uptake, storage, and oxidation. Collectively, intestinal microbiota exert an intrinsic impact on skeletal muscle growth, development, and metabolism. Thus, the underlying relationship between skeletal muscle and gut microbiota merits future investigation to produce high quality pork.

## 3. The Role of Gut Microbiota in Fat Deposition

Globally, the incidence of obesity and obesity-related complications is increasing quickly [44], which has attracted more and more attention. Consequently, people are often consciously controlling lipids intake for health [45]. Simultaneously, excessive fat accumulation induces a huge waste of dietary energy in animals, severely affecting economic returns in animal husbandry [13]. Accordingly, it is significant to investigate the factors affecting porcine fat deposition. Indeed, porcine fat accumulation is affected by many factors, including genetics, feed, and management. Nowadays, cumulating evidence in humans and mice points strongly to the relationship between gut microbiota and host fat accumulation [9,15]. Furthermore, the gut microbiota, including *Lachnospiraceae*, *Ruminococcaceae*, *Prevotella*, *Treponema*, and *Bacteroides*, have been shown to significantly influence fatness in pigs [13,46]. Therefore, it may be possible to develop novel strategies to regulate the porcine fat accumulation by altering intestinal microbiota, and the underlying mechanisms linking intestinal microbiota and fat accumulation is warranted to further investigation, as summarized in Figure 1.

### 3.1. Gut Microbiota Metabolites

Gut microbiota are essential to host health and well-being [47], depending heavily on the microbial metabolism [48]. Indeed, mounting evidence suggests that metabolites derived from gut microbiota, e.g., SCFAs, amino acids and their derivatives, and bile acids (BAs) and their derivatives, have been implicated in modulating host lipid metabolism [49].

SCFAs are the ligand of G protein-coupled receptors 43 (GPR43) or GRP41 on enteroendocrine cells [30], and its combination with GPR43 or GRP41 can induce the secretion of glucagon-like peptide-1 (GLP-1) or peptide tyrosine tyrosine (PYY), respectively [50], thus indirectly affecting satiety and insulin secretion, and eventually regulating body fat deposition. SCFAs have also been suggested to be involved in pig lipid metabolism. Specifically, Jiao et al. demonstrated that SCFAs treatment could suppress fat accumulation by the inhibition of lipogenesis and promoting lipolysis of different tissues in weaned pigs [51,52]. Similarly, SCFA exogenously introduced into the ileum regulates expression levels of lipogenesis and lipolysis related genes in the longissimus dorsi muscle, abdominal fat, and liver of growing pigs, improving meat quality [53]. These studies give novel insights into the action of SCFA in swine production as feed additives to achieve high-quality pork production.

It was reported that the ability to produce branched-chain amino acids (BCAAs, including valine, leucine, and isoleucine) and aromatic amino acids (AAAs, including phenylalanine, tyrosine, and tryptophan) by gut microbiota of obese individuals was higher than that of lean controls [54], implying a correlation between these gut microbiota-derived amino acid and lipid metabolism. In addition, Chen et al. revealed that *Prevotella copri* triggered chronic inflammation through promoting the production of several metabolites (LPS, BCAA, AAA, and arachidonic acid), resulting in porcine fat deposition [13]. In pigs, addition of BCAA into a low-protein diet alters body fat condition; however, the change of skeletal muscles [55], liver [56], subcutaneous adipose, abdominal subcutaneous adipose, and perirenal adipose [57,58] are different, perhaps as a result of the expression of regulators of lipid metabolism in these tissues in a depot-specific manner. The gut microbiota can convert tryptophan into indole and related derivatives, e.g., indole-3-aldehyde, indole-3-acid-acetic, indole-3-propionic acid, indole-3-acetaldehyde, and indole acrylic acid [59], some of which serve as endogenous ligands of the aryl hydrocarbon receptor (AhR). In addition, previous studies indicated that the AhR performs essential roles in regulating lipid or fatty acid metabolism [60]. Similarly, evidence in mice and rats suggested tryptophan and its derivatives can affect lipid metabolism [61,62]. In animal production, as a feed additive, tryptophan can regulate animal lipid metabolism and improve animal growth performance and product quality. For example, Wang et al. found that dietary supplementation reduced the abdominal fat content of male broilers [63]. The above observation indicates that BCAA or AAAs supplementation exert effects on lipid metabolism; however, additional research is needed to systematically clarify the role of gut microbiota.

The primary bile acids (BAs) are synthesized from cholesterol and essential for lipid/vitamin digestion and absorption, and are subsequently converted into secondary bile acids by microbial deconjugation, dehydrogenation, dihydroxylation, and epimerization [64]. Bacterial bile salt hydrolase (BSH), a deconjugation enzyme, lies in many bacteria, including *Bacteroides thetaiotaomicron*, *Bifidobacterium longum*, *Enteroccocus faecalis*, *Ligilactobacillus salivarius,* and *Clostridium perfringens* [65]. Critically, Joyce et al. demonstrated that bacterial BSH activity significantly affects the lipid metabolic processes and adiposity in the host [66], which indicates indirectly that gut microbiota is vital for lipid metabolism. Additionally, Bas can serve as important signaling molecules that regulate host lipid metabolism by activating the cellular farnesoid X receptor (FXR) or G protein-coupled receptor 5 (TGR5) [67,68]. Indeed, a previous study in mice indicated that microbiota cause weight gain, steatosis, and inflammation by activating FXR signaling [69]. However, future investigation is still required to elucidate whether gut microbiota affect host metabolism via TGR5.

### 3.2. Microbiota–Gut–Brain Axis in Fat Deposition

It is well known that several avenues, including the immune system, direct enteric nervous system routes and the vagus nerve, can be a bridge of bidirectional communication between the microbiota and the brain [70]. They can integrate the distal and local regulatory networks to regulate many physiological processes of the host, then influence the overall metabolism.

Metabolites derived from gut microbiota can modulate brain function by the vagus nerve (VN), blood–brain barrier (BBB), and immune system [70,71]. Notably, the vagus nerve is important for gut microbiota–brain crosstalk [72], as inferred by studies on gut microbiota, including *Bifidobacterium longum*, *Lactobacillus rhamnosus*, and *Limosilactobacillus reuteri*. However, these effects are blunted after vagotomy [73,74,75,76]. SCFAs control the secretion of gut hormones from enteroendocrine cells (EECs), including PYY, GLP-1, cholecystokinin (CCK), ghrelin, and leptin, which affect the vagal afferent pathway and thus regulate host ingestion, energy balance, and circadian rhythm [77,78]. In addition, evidence also suggests that SCFAs contribute to the inhibition of food intake via the activation of VN [79]. Moreover, evidence from high fat fed rats showed that increased acetate production can activate the parasympathetic nervous promoting an increased secretion of glucose-stimulated insulin and ghrelin, which induces liver and muscle insulin resistance, hyperphagia, hypertriglyceridemia, and ectopic lipid deposition within liver and skeletal muscle [80]. Apart from gut hormones, the peripheral terminals of vagal afferent also respond to lipopolysaccharide (LPS). Specifically, a previous study suggested that chronic exposure to low-dose LPS caused leptin resistance of vagal afferent neurons and reduced sensitivity to CCK-induced satiation, leading to the loss of vagal afferent plasticity [81]. This demonstrated that the change of microbiota may contribute to hyperphagia and further result in obesity. Interestingly, monocarboxylate transporters are located on the endothelial cells of the BBB, thus SCFAs are able to cross the BBB [82], indicating critical roles of BBB in gut microbiota–brain cross talk. Indeed, after intraperitoneal injection of acetate, decrease in food intake has been associated with acetate entering the brain over the BBB and directly activating hypothalamic neurons [83]. Importantly, gut microbiota are intimately associated with the maturation and function of microglia, which are the main neuroimmune cells [71,84]. Specifically, in vivo studies using germ-free mice or antibiotic-treated mice found that intestinal microbiota disorder caused microglia with an immature phenotype [85]. These studies showed that an intimate connection exists between gut microbial metabolites and brain inflammation. In addition, recently, hypothalamic inflammation has been demonstrated to break the energy homeostasis, thereby leading to development of glucose intolerance, insulin resistance, and obesity [86]. Nevertheless, the pathogenesis of hypothalamic inflammation has not been fully understood, the metabolites derived from gut microbiota may be involved.

Collectively, the study of microbiota–gut–brain axis is of great significance to the regulation of nutrient metabolism in animals. Dietary behavior controlled by the central system can be regulated through the microbiota–gut–brain axis to affect the body lipid metabolism. In addition, the regulation of the microbiota–gut–brain axis provides ideas to cure metabolic diseases (such as obesity) or to improve animal fat deposition related to meat quality.

### 3.3. Fecal Microbiota Transplantation (FMT)

FMT has been used to treat multiple diseases in humans including obesity related complications [87]. FMT is an effective and promising therapy, as the entire gut microbiota and metabolites can be transferred from donor to the recipient. 

Interestingly, FMT from obese humans or mice into germ-free or antibiotic treated mice leads to body fat accumulation [88,89], implying that FMT can be used as a means to investigate the impact of gut microbiota on fat deposition. Mounting evidence suggests that the composition of the intestinal microbiota is different between obese and lean pigs [4,26]. For instance, obese Rongchang pigs presented a remarkably higher Firmicutes/Bacteroidetes proportion and obvious genera differences compared with lean Yorkshire pigs [26]. Xiao et al. also revealed that bacterial taxa were different in obese Jinhua and lean Landrace pigs [90]. Nevertheless, the extent of the contribution of gut microbiota to porcine lipogenic characteristics are still unclear. Accordingly, many studies further explored the mechanism of gut microbiota affecting fat deposition via FMT. Pigs-to-germ free mice FMT has provided evidence for the fact that the gut microbiota of pigs inherently influence lipid metabolic profiles and fat deposition. Yan et al. indicated that Rongchang pigs and their recipients (germ-free mice) displayed higher body fat mass and stronger lipogenesis in the gastrocnemius muscle than Yorkshire pigs and their germ-free mouse recipients [26]. In line with the above observation, the mice receiving Jinhua pigs’ “obese” microbiota had increased lipid and triglyceride levels and the lipoprotein lipase activity in the liver relative to those receiving Landrace pigs’ “lean” microbiota [4]. These studies demonstrate that lipogenic properties from pigs are moved to germ free mice by FMT. Although its potential is exciting, there are obstacles to the use of FMT in practical applications. Factors restricting wider application of FMT include problems with donor selection, a lack of optimized methods for the preparation of the FMT, recipient genetics, lifestyle, and microbiota composition. Indeed, both microbial diversity and the presence of specific species in the recipient microbiota have been suggested to affect FMT engraftment [91]. Thus, FMT efficacy is highly linked with donors, appropriate FMT protocol, and recipient clinical status.

In conclusion, gut microbiota contribute to the fat deposition in pigs, and their fat deposition phenotype can be transferred across species. Therefore, transplantation of gut microbiota will provide new approaches to change host metabolism and adipogenesis, with a huge significance in the improvement of meat quality.

## 4. Factors Affecting Gut Microbial Composition and Function

### 4.1. Host Genetics

Host genetics are crucial to shape the composition gut microbiota [92]. Mounting evidence also suggests that some bacterial taxa were identified as heritable in human and pigs [92,93,94,95,96,97]. In pigs, previous reports suggest that the intestinal microbiota from obese and lean pigs are not exactly the same [4,26,90]. The existance of different core microbiota has been confirmed between Rongchang pigs (obese) and Yorkshire pigs (lean) by high-throughput pyrosequencing [26]. Additionally, the first four genera in Jinhua pigs (obese) are different from those in Landrace pigs (lean) [4]. A recent study also suggested that obese Shaziling pigs possess a higher α-diversity and higher abundances of probiotics, including *Lactobacillus amylovorus*, *Lactobacillus johnsonii*, and *Clostridium butyricum*, compared with Yorkshire pigs [7]. Interestingly, when different purebred pigs were randomly assigned to cohouse for several weeks, their gut microbial communities became more similar, while the distinguishable breed-particular proportions were retained [98], which imply that host genetics play a causal role in gut microbiota differences. Furthermore, a total of 52 single-nucleotide polymorphisms (SNPs) distributed in 17 regions along the pig genome were associated with the relative abundance of six genera, including *Akkermansia*, *CF231*, *Phascolarctobacterium*, *Prevotella*, *SMB53*, and *Streptococcus* [99]. Recently, Reverter et al. identified SNPs with pleiotropic effect associated with the relative abundance of butyrate producer bacteria (*Faecalibacterium*, *Butyrococcus*, and *Coprococcus*) and identified candidate genes previously reported as associated with microbiota profile in mice and human, such as *SLIT3*, *SLC39A8*, *NOS1*, *IL1R2*, *DAB1*, *TOX3*, *SPP1*, *THSD7B*, *ELF2*, *PIANP*, *A2ML1*, and *IFNAR1* [100]. Moreover, a study in two herds of pigs which were kept on same farm conditions and fed with the same diet identified tens of heritable bacterial taxa by heritability estimates and 37 candidate genes associated with gut microbiota. Interestingly, 13 out of these 37 candidate genes are associated with metabolism, including circulating glucose and insulin level and obesity [97]. Consequently, the effect of gut microbiota on fat deposition in obese and lean pigs may be regulated by host genetics. 

Such data confirm the significance of host genomics in regulating the microbiota composition. This area merits further research to understand the complicated interplay between the host genotype and gut microbiota taxa, identifying genetic markers and candidate genes that can be used to incorporate in pig breeding program to reshape microbial composition, and thus improving host performance. 

### 4.2. Diets

However, previous study suggested that host genetics play a secondary role in shaping microbiota composition, perhaps because conditions are difficult to standardize between individuals. Recently, a study showed that environment dominates over host genetics in determining host gut microbiota [101]. Among environment factors, dietary factors, a powerful shaper of microbiota composition, rapidly and reproducibly alter gut microbial community structure [14]. In addition, diet can improve pig performance and pork quality via modulating gut microbial diversity and composition [102,103]. 

Importantly, dietary fat quantity has been reported to consistently influence gut microbiota composition. For instance, a large-scale meta-analysis of sequencing-based studies suggested from rodents and humans that a high-fat diet (HFD) influence microbial diversity and reproducibly changes gut microbial community structure, including the increase in Firmicutes/Bacteroidetes ratio, which was significantly correlated with fat content [104]. This may be the reason why pigs become obese after consuming HFD [105]. Further, Kong et al. suggested that the consumption of a HFD has been shown to reduce several beneficial bacteria, e.g., *Lactobacillus*, *Prevotella*, *Alloprevotella*, and *Clostridium sensu stricto*, but increase opportunistic pathogen including *Bacteroides*, *Alistipes*, and *Anaerotruncus* [106]. Therefore, when designing pig feed formula, fat should be added appropriately to maintain the balance of gut microbiota. In addition to dietary fat quantity, the consumption of different dietary fat type differently shifts gut microbial composition and function [107,108]. Currently, SCFAs and middle chain fatty acids (MCFA) are known to have antimicrobial properties. For example, diet with such lipids inhibit the growth of common causative agents (*Salmonella* and *E. coli*), thereby reducing the odds of gastrointestinal infections [109,110]. Moreover, there are three main types of fat in mammals diet based on saturation: saturated fatty acids (SFA), monounsaturated fatty acids (MUFA), and polyunsaturated fatty acids (PUFA), which potentially regulate pig gut microbiota composition. For instance, pigs fed palm oil had marked changes in the bacteria community structure through raising the abundance of *Proteobacteria* as well as decreasing the abundance of *Firmicutes* [108]. Additionally, pigs fed with oleic acid had higher abundance of *Prevotella* [111]. Moreover, a flaxseed oil supplemented diet may enhance intrauterine growth retardation in pigs’ gut immunity and health, which was attributed to the changed gut microbiota and mucosal fatty acid profile [112]. Together, dietary fat types are closely related to gut microbiota structure, offering a novel insight into the reasonable application of fat in diet.

In addition, recently, due to the critical role in shaping gut microbiota, dietary fiber has increasingly attracted the attention of researchers. Dietary fiber, a feed component that cannot be digested effectively by monogastric digestive enzymes, serves as the major energy source for gut microbiota, which implies that intake of the right amount of dietary fiber could enhance the abundance of specific microbiota [113]. As recently reviewed by Pu et al. [102], adding fiber to a diet not only promotes the increase in the a few fiber-degrading bacteria taxa, but also the functional activity of the microbiota and SCFA in finishing pigs. In addition, they found that elevating the proportion of fiber in the diet increases the α-diversity and β-diversity indices of the porcine gut microbiota [102]. Importantly, dietary fiber supplementation not only upregulates the proportion of beneficial bacteria but also diminishes the proportion of pathogenic bacteria in the gut [114,115,116], which may be conducive to the improvement of intestinal health and meat quality of pigs [117,118]. Nevertheless, dietary fiber from different sources has been investigated to have different effects on microbial composition and diversity. Indeed, a previous study showed that xylan is beneficial for the proliferation of Bifidobacterium, whereas glucan is unfavorable for it in the ileum and cecum of pigs [119]. Moreover, a study reported that alfalfa diets increase the relative abundance of *Firmicutes* and *Bacteroidetes* in weaned piglets compared with wheat bran [120]. The above research studies show a close relationship between dietary fiber and gut microbiota, yet the specific mechanism of dietary fiber affecting porcine gut microbiota needs to be further studied.

Collectively, the available evidence suggests that manipulation of the dietary composition can influence the gut microbiota structure and function in pigs and thus could contribute a key role to the pig growth and development, and thus, reasonable diet formula is conductive to the production of better pork in pig industry.

### 4.3. Antibiotic

In intensive swine husbandry systems, antibiotics are often added to porcine feed or water to fight against the infections of the respiratory system and gastro-intestinal tract, and promote pig’s growth and development. In addition, the use of different types of antibiotics is associated with specific diseases, age, or farm management. For example, penicillins were widely used for prophylaxis and treatment of septicaemia and respiratory and urinary tract infections in various animal species. Tetracyclines were commonly used to treat respiratory diseases caused by *Actinobacillus pleuropneumonia* and *Pasteurella multocida*. Colistin was most commonly used in gastrointestinal conditions of piglets and weaners, while tylosin was used in fatteners and sows [121]. However, large volumes of antibiotics used in food animals contribute to the emergence and spread of antimicrobial resistant (AMR) that lie in the food chain to propagate to consumer [122]. Thus, a detailed understanding of the current pattern of antibiotic use in livestock is essential to support optimal antibiotic use. This could decelerate the emergence of AMR in animal production. Additionally, renewed attention has been dedicated the association between antibiotic exposure and host physiology, acknowledging the critical role of gut microbiota in host physiology deciphered in recent years. A study reported that antibiotic treatment influences porcine growth performance, myofiber composition, and lipid metabolism, which could be related to alteration of gut microbiota [123]. However, long-term use of subtherapeutic doses of antibiotics can increase the proportions of pathogenic microorganisms that inhibit the normal intestinal function, leading to an adverse impact on the commensal bacterial population and triggering diseases [124,125,126]. Specifically, long-term exposure to antibiotics may result in the generation of intestinal pathogens and postweaning diarrhea by reducing bacterial diversity and stimulating gut inflammation [127]. Studies in piglet showed that the addition of subtherapeutic antibiotic from postnatal day 7 to 42 differently altered the microbial composition in the small and large intestines, e.g., a reduction in *Clostridium*, *Bacillus*, and *Sharpea* in the digesta of the stomach, duodenum, and jejunum as well as a minor decrease in *Prevotella* in colon, thus affecting metabolic phenotype [128,129]. Several antibiotics, including penicillin, tylosin, sulfamethazine, and chlortetracycline, have been demonstrated to alter markedly the gut microbiota composition and thus break the balance in growing pigs [130,131,132]. However, the number of strains resistant to antibiotics are increasingly found in swine production [133]. Additionally, antibiotic use in early life could interfere with the precedence effect of early colonizers, slow down normal maturation of gut microbiota, and possess a long-lasting effect on intestinal microbiota [134,135]. As Schokker et al. reported, piglets that received a subcutaneous injection with tulathromycin on day 4 after birth exhibited a reduction in the microbial diversity and a change microbial composition in jejunum on day 176, but had limited effect on day 56 [136]. The dysbiosis can influence host the nutrient absorption and utilization, as well as life-long metabolism phenotype and diseases, including obesity [137,138,139,140]. Due to the increasing safety concerns of antibiotics, the use of antibiotics in animal feed have been restricted in many countries. Consequently, understanding the post-term effect and specific mode of action of the antibiotics is of great importance to better design antibiotic alternatives in pigs.

### 4.4. Prebiotics and Probiotics

Prebiotics and probiotics have been applied as functional feed additives to improve animal health and production performance for many years. Prebiotics were defined as “a selectively fermented ingredient that allows specific changes, both in the composition and/or activity in the gastrointestinal microflora, thus conferring benefits upon host wellbeing and health” [141]. So, prebiotics have the ability to enhance the growth of beneficial resident gut bacteria. Well-known effective prebiotics include inulin, galacto-oligosaccharides, xylo-oligosaccharides, pectin, beta-glucans, and resistant starch, and supplementation of these prebiotics in porcine diet has been confirmed to improve microbiota composition and functionality, and is beneficial for intestinal barrier functions and growth performance (Table 1). From these observations, the benefits of prebiotics are usually attributable to the following aspects: (a) stimulating beneficial bacteria and SCFAs production and, accordingly, improving barrier function, and anti-inflammatory stimuli; and (b) decreasing levels of some pathogenic organisms (such as *Defluviicoccus* and *Gardnerella*) that cause gut dysbiosis. Additionally, prebiotics can modulate lipid metabolism and result in a higher IMF level in finishing pigs [142], possibly by inhibiting lipogenic enzymes activity, and subsequently reducing production of lipoproteins and TGs [143]. In brief, the action of prebiotics is exerted via altering the certain bacteria abundance and the metabolites that are produced.

Probiotics applied in animal diets are predominantly bacterial strains of Gram-positive bacteria, and are already widely used for replenishing the gut microbial population and modulate gut microbiota composition imbalance [154], while restoring host immune system and for weight gain in farm animals [155]. Highly diverse organisms of the genera *Lactobacillus*, *Bacillus*, *Enterococcus*, and *Pediococcus* are already widely used in swine production [156]. Owing to the main goals of applying probiotic vary by growth phase, appropriate probiotics strains should be selected at different stages of life. The use of probiotics for weaning piglets has been widely reported in the literature. For example, many studies demonstrated that probiotics, such as *lactic acid bacteria* and *Saccharomyces cerevisiae*, can prevent ETEC K88/F4 colonization [157,158,159], which enhance immunity, balance the environment in the gastrointestinal tract and decrease the occurrence of diarrhea, thus improving production performance. In addition, *L. rhamnosus*, *L. plantarum,* and *Bacillus* has been reported to clear pathogens via triggering host immune response [160,161,162]. During the growing–finishing stage, pigs get more protection from a mature immune system, so the purpose of using probiotics is to improve growth performance and produce better pork. Bacillus subtilis, *L. plantarum*, *L. acidophilus* NCDC-15, and *P. acidilactici* FT28 exhibited beneficial impact on growth performance of grower–finisher pigs [163]. Samolińska et al. indicated that a multispecies probiotic preparation supplementation has a positive on fattening performance and health status of growing pigs [164]. In conclusion, prebiotics and probiotics are microbiota-management tools for improving pig gut health, energy metabolism, and growth performance, which are closely associated with meat quality.

## 5. Conclusions

It is generally acknowledged that there is a complex relationship between gut microbiota and the pig, suggesting that microbiota modifications induce alternations in the pig growth and metabolism, which potentially contribute to changes in fiber characteristics, fiber type distribution, and lipid metabolism in skeletal muscle, and regulation of fat deposition. Hence, factors affecting gut microbiota composition and function have taken a pivotal role in pig production (Figure 2).

The gastrointestinal tract of pigs harbors a rich and highly complicated microbiota, affecting digestion and absorption of nutrients; however, the composition and function of the gut microbiota are dynamic and regulated by many factors, such as host genetics, diet properties, antibiotic, prebiotics, and probiotics. Dietary compositions have been shown to significantly affect gut microbiota composition in terms of richness and diversity. On one hand, consumption of HFD (particularly saturated fat) may induce the growth of pathogenic bacteria, while suppressing the growth and reproduction of beneficial bacteria, resulting in gut microbiota dysbiosis. On the other hand, the consumption of different dietary fat type has been proven to differently change gut microbial composition and function. Moreover, the consumption of dietary fiber may increase the number of the of beneficial bacteria, promoting SCFAs production and subsequently affecting energy metabolism. Although antibiotic administration reduced pathogens colonization, the extensive and long-term use of antibiotics caused the generation of resistant bacteria and antibiotic residue in pigs. Therefore, alternative sources to antibiotics are urgently need to discover promoting the development of pig industry. More recently, prebiotics and probiotics, which reinforce the gut microbiota for improved animal health and colonization resistance to gut pathogens, show promise as alternatives for antibiotics for growth promotion in swine industry. Therefore, modifying the gut microbiota via dietary interventions is a vital strategy to improve the health and meat quality of the pig.

Future investigation on the association between specific gut microbiota and pig skeletal muscle development, fat deposition, and further exploration of factors affecting gut microbiota composition, structure, and function would be beneficial to better understand the role of gut microbiota in pigs, which would contribute to the production of better pork.

## Figures and Tables

**Figure 1 antibiotics-11-00793-f001:**
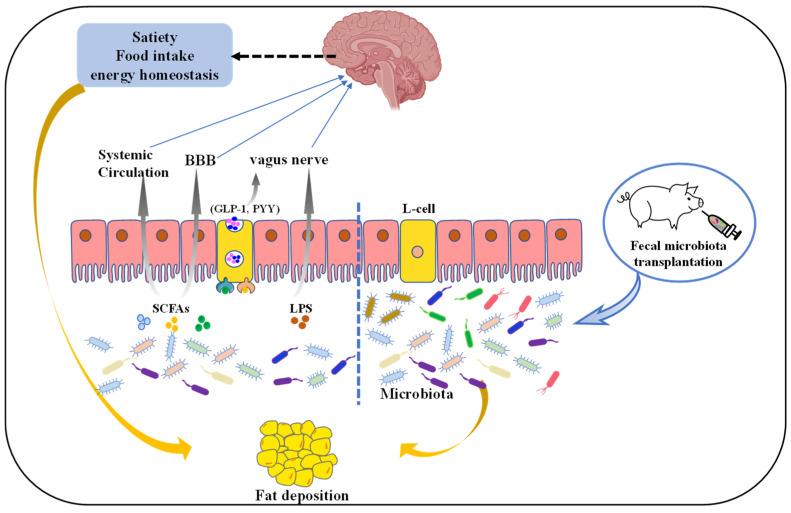
The mechanism of gut microbiota regulates pig fat deposition. Microbiota-derived metabolites such as SCFAs or lipopolysaccharide (LPS) affect satiety, food intake, or energy homeostasis directly via the vagus nerve (VN) or indirectly blood–brain barrier (BBB) and systemic circulation. L-cells are activated by these microbial metabolites through activation of different receptors, resulting in the production of gut hormones such as glucagon-like peptide-1 (GLP-1) and peptide YY (PYY). These intestinal hormones signal from the gut to the nucleus tractus solitarius in the brain via the VN and direct secretion into the circulatory system. Additionally, fecal microbiota transplantation (FMT), where fecal microbiota from a donor is transplanted into a recipient GI tract, has the potential to be an effective option to regulate fat deposition.

**Figure 2 antibiotics-11-00793-f002:**
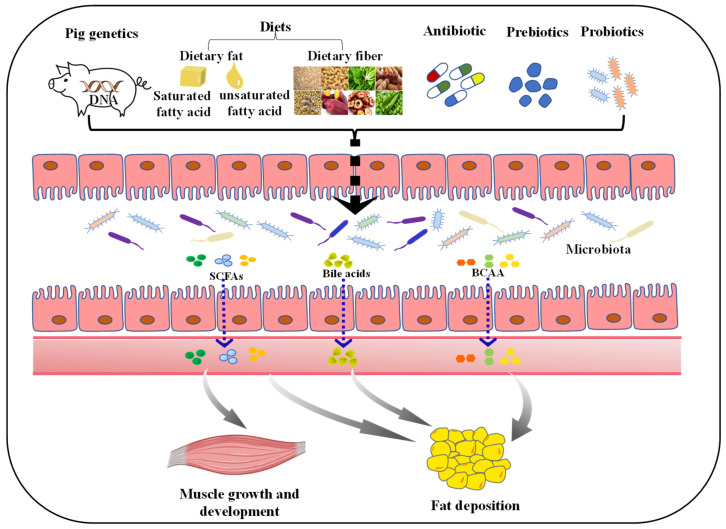
Factors affecting porcine muscle growth and development, and fat deposition via regulating gut microbiota. The diversity of gut microbiota can be influenced by many factors. Both host genetics and diet composition (such as dietary fat, fiber) play a pivotal role in shaping the gut microbiota. Moreover, subtherapeutic doses of antibiotics applied in commercial farming can negatively impact the microbiota. The addition of probiotics and prebiotics could improve gut microbiota diversity. Ultimately, these factors can affect the growth and development of muscle, and fat deposition in pigs by regulating the composition and metabolites (such as SCFAs, bile acids and BCAA) of gut microbiota. SCFAs, short-chain fatty acids; BCAA, branched-chain amino acids.

**Table 1 antibiotics-11-00793-t001:** Effects of different prebiotics on gut microbiota and their impact on host health.

Dietary Sources of Prebiotics	Study Model	Effect on Gut Microbiota	Effect on Host Health	Effect on Growth Performance	Reference
A diet supplemented with 5% microcrystalline cellulose or inulin for 72 days	Pig	Alter the composition of ileal and colonic mucosal microbiota ↑*Akkermansia* and ↑production of butyrate	Enhance sulfomucin production and mucosal barrier function	Not mentioned	[144]
A basal diet containing 2.5, 5.0, and 10.0 g/kg inulin	Thirty-two male weaned pigs	↑Lactobacillus population ↓Escherichia coli population in the caecum ↑Production of acetic and butyric acid	Elevate serum insulin-like growth factor-1 concentration but reduced diamine oxidase concentration;	2.5 g/kg inulin increase the average daily feed intake (ADFI) and average daily body weight gain (ADG) of pigs	[145]
A basal diet containing 0.5% inulin for 21 days	Twenty growing-pigs	↑*Lactobacillus* spp. in the ileum and ↑*Bacteroides* spp. in the cecum ↑Production of acetate and butyrate in cecum	Increase villus height and the abundance of zonula occludens-1; Decrease IL-6 and TNFαexpression, and reduce gut epithelial cell apoptosis in ileum and cecum	Not mentioned	[146]
The combined of the early-life galacto-oligosaccharides (GOS) and postweaning GOS intervention	Weaning Piglets	↑ The abundances of (SCFA) producers ↑Total SCFA concentration	Reduce the expression of MyD88-NF-κB signaling and the proinflammatory cytokines	Not mentioned	[147]
Piglets in the GOS group were given 10 mL of GOS solution daily	Neonatal Porcine Model	↑*Lactobacillus* and unclassified *Lactobacillaceae*, ↓*Clostridium sensu stricto* on day 8 and day 21 after GOS intervention. ↓*Escherichia* on day 21 following the early-life GOS intervention	Increase microbial metabolites (such as SCFAs, and Lactate), endocrine peptides, and the mRNA expression of anti-inflammatory cytokines and antimicrobial peptides	Not mentioned	[148]
xylo-oligosaccharides group (basal diet + 250 g t-1 XOS)	weaned piglets	↑The concentrations of butyrate in the ileum and tryptamine and spermidine in the colon ↓The concentration of indole in the colon	Improve the growth performance	Increase the ADFI and ADG	[149]
Piglets were fed a low-methyl esterified pectin enriched diet, a high-methyl esterified pectin enriched diet, a hydrothermal treated soybean meal enriched diet or a control diet	weaning pigs	↓Abundance of the genus *Lactobacillus* ↑Abundance of *Prevotella*	Affect the digestion processes; Shape the colonic microbiota from a Lactobacillus-dominating flora to a *Prevotella*-dominating community, with potential health-promoting effects	Not mentioned	[150]
piglets were fed with yeast-derived β-glucans	piglets	↓Abundance of *Methanobrevibacter* ↑Abundance of genera *Fusobacterium* and *Ruminococcaceae**_UCG-002*	Did not affect the vaccination response; Affect modestly fecal microbiota composition and immune parameters	Not mentioned	[151]
Pigs received a diet amended with 5% resistant potato starch	piglets	↑anaerobic *Clostridia* and ↑Production of butyrate	Increase the abundance of regulatory T cells in the cecum; Modulate the microbiota and host immune status, altering markers of cecal barrier function and immunological tolerance	Not mentioned	[152]
Piglets fed a diet with 0.5% PD	weaned piglets	↓Abundance of pathogenic organisms, such as *Defluviicoccus* and *Gardnerella* ↑*Psychrobacter* and *Prevotella* ↑SCFA-producing bacteria	Increase the concentration of SCFAs in the feces	At 42 days of age, dietary PD supplementation increase the body weight (P = 0.06); Increase the feed efficiency	[153]

↑, increased; ↓, decreased.

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
