# Peer review of "The Role of Gut Microbiota in the Skeletal Muscle Development and Fat Deposition in Pigs"

_antibiotics, 2022, doi:10.3390/antibiotics11060793_

Round 1
Reviewer 1 Report
The manuscript reviewed the role of gut microbiota in pigs for skeletal muscle development and fat deposition. The topic addressed in this study falls within the scope of Antibiotics. Generally, the review was well conceived and well presented. The manuscript contains useful information with room for practical application in the pig industry. However, there are few corrections that need to be done to improve the quality of the manuscript. Thus, I recommend the acceptance of the manuscript for publication subject to a minor revision. The corrections are as follows
Line 14: Delete ‘people having perform
Line 16: Replace ‘more and more’ with ‘increased
Line 19: Replace ‘all will be’ with ‘were also
Line 21: Insert ‘of’ after ‘understanding
Line 22: Replace ‘will contribute to produce’ to ‘may contribute to the production
Line 68: Replace ‘the reduction’ with ‘a reduction
Line 83: Cite the reference ‘ Qi et al’ properly as per Vancouver style
Line 191: Insert ‘that’ after ‘than
Line 349: Delete ‘to prove
Line 356-357: Replace ‘show richer in’ with ‘had higher abundance of
Line 358: Insert ‘of’ before ‘pigs
Line 358: Replace ‘attribute’ with ‘was attributed to
Line 362: Delete ‘been
Line 367: Cite the reference ‘ Pu et al’ properly as per Vancouver style
Line 367: Please check and correct ‘in the a
Line 379: Replace ‘relative’ with ‘relationship
Line 388: Delete ‘to
Line 395: Insert ‘to’ before ‘an
Line 402: Please check ‘a small drop weekly
Line 467: Delete ‘on'
Author Response
Reviewer #1:
Comments and Suggestions for Authors
The manuscript reviewed the role of gut microbiota in pigs for skeletal muscle development and fat deposition. The topic addressed in this study falls within the scope of Antibiotics. Generally, the review was well conceived and well presented. The manuscript contains useful information with room for practical application in the pig industry. However, there are few corrections that need to be done to improve the quality of the manuscript. Thus, I recommend the acceptance of the manuscript for publication subject to a minor revision. The corrections are as follows
Line 14: Delete ‘people having perform
Response: Thank you for your review, we have deleted “people having perform” according to your comments. (Revised manuscript: Line 16)
Line 16: Replace ‘more and more’ with ‘increased
Response: Thank you for your advice, we have replaced “more and more” with “increased”. (Revised manuscript: Line 18)
Line 19: Replace ‘all will be’ with ‘were also
Response: Thank you for your comment. We have made this revision. (Revised manuscript: Line 21)
Line 21: Insert ‘of’ after ‘understanding
Response: Thank you for your suggestion. We have inserted “of” after “understanding”. (Revised manuscript: Line 23)
Line 22: Replace ‘will contribute to produce’ to ‘may contribute to the production
Response: Thank you for your comment. We have made this revision. (Revised manuscript: Line 24)
Line 68: Replace ‘the reduction’ with ‘a reduction
Response: Thank you for your comment. We have replaced “the reduction” after “a reduction”. (Revised manuscript: Line 70)
Line 83: Cite the reference ‘ Qi et al’ properly as per Vancouver style
Response: Thank you for your review, we have made this revision. (Revised manuscript: Line 85)
Line 191: Insert ‘that’ after ‘than
Response: Thanks. We have inserted “that” after “than”. (Revised manuscript: Line 194)
Line 349: Delete ‘to prove
Response: Thanks. We have deleted “to prove”. (Revised manuscript: Line 357)
Line 356-357: Replace ‘show richer in’ with ‘had higher abundance of
Response: Thank you for your suggestion. We have replaced “show richer in” after “had higher abundance of”. (Revised manuscript: Line 366)
Line 358: Insert ‘of’ before ‘pigs
Response: Thank you for your comment. We have inserted “of” before “pigs”. (Revised manuscript: Line 367)
Line 358: Replace ‘attribute’ with ‘was attributed to
Response: Thank you for your suggestion. We have replaced “attribute” after “was attributed to”. (Revised manuscript: Line 367-368)
Line 362: Delete ‘been
Response: Thanks. We have deleted “been”. (Revised manuscript: Line371)
Line 367: Cite the reference ‘ Pu et al’ properly as per Vancouver style
Response: Thank you for your review, we have revised the issue. (Revised manuscript: Line 376)
Line 367: Please check and correct ‘in the a
Response: Thank you for your suggestion. We have corrected “in the a”. (Revised manuscript: Line 376)
Line 379: Replace ‘relative’ with ‘relationship
Response: Thank you for your suggestion. We have replaced “relative” with “relationship”. (Revised manuscript: Line 389)
Line 388: Delete ‘to
Response: Thank you for your comment. We have deleted “to”. (Revised manuscript: Line 409)
Line 395: Insert ‘to’ before ‘an
Response: Thank you for your suggestion. We have inserted “to” before “an”. (Revised manuscript: Line 415)
Line 402: Please check ‘a small drop weekly
Response: Thank you for your comment. We have replaced “a small drop weekly” with “a minor decrease”. (Revised manuscript: Line 422)
Line 467: Delete ‘on'
Thank you for your suggestion. We have deleted “on”. (Revised manuscript: Line 487)
468

Reviewer 2 Report
The authors aim to synthesise across relevant previous studies on pig gut microbiota, and fat and muscle development to discuss how microbiota may affect pork quality for human consumption. Consumers are becoming more concerned about the health of the animal whose meat they are consuming. Particularly, there are concerns revolving around antibiotic and steroid consumption in animals that have carry-over effects in human consumers. In the changing consumer market and consumer behaviour landscape, I believe this review may be of importance and serve as an encouraging stepping stone for studies that aim to ameliorate meat quality through both better animal husbandry and improved human health outcomes.
General comments:
I suggest the title is changed to: "The role of gut microbiota in the skeletal muscle development and fat deposition in pigs"
I would like to see the authors discuss what type of practises are currently used in the animal husbandry that are actually harming the health of pigs and their human consumers. E.g., historically, antibiotics have been used pre-emptively is animal feed to "proactively fight bacterial infections" (dangerous due to antimicrobial resistant (AMR) strain increase) and to increase body mass of the animal to produce more meat. This is an important part of the discussion and should be included in the review. I would also like to read more about how the different types of antibiotics affect the many positive effects of an intact and natural gut microbiota the farm pigs already have. I believe this part is vital considering the journal is called Antibiotics and the current part on antibiotics is too brief. Authors could also make note in the discussion of how AMR in pigs could serve as a dangerous source of zoonotic diseases
The discussion should focus on how to improve current practises using the knowledge we have at the moment by providing example strategies for future
Authors need to proofread that all bacterial species and strain names are in italics
When discussing FMT, the authors should also indicate the potential cons of FMT, e.g., how it may disturb the co-adapted microbiota, how engraftment is usually imperfect, and how the FMT-introduced bacteria will oftentimes be outcompeted by local strains after a while
Considering how some breeds of pigs seem to have distinct gut microbiota that makes them more prone to obesity, it would be wise to discuss how artificial selection by humans for higher body mass and fat content in pigs has likely had a role in this. Role of domestication and breeding has not been discussed by the authors
Authors discuss "beneficial bacteria" when talking about pre- and probiotics. Are these beneficial for the pigs (will they be actually healthier?) or are they beneficial for us (we like a certain pork quality and these bacteria will give us the desired effect?)
Figure 1 and 2 are in the wrong order. The first figure is indicated as Figure 2 and vice versa
Specific edits:
Line 18: change to "purpose is"
Line 19: change to "the factors affecting gut microbiota composition will also be discussed"
Line 21: change to "understanding between the relationship between gut microbiota and meat quality in pigs, and how manipulation of gut microbiota may contribute to increasing pork quality for human consumption"
Line 26: remove "as"
Line 30: I suggest the authors make a comparison between pork consumption and other meat consumption. I.e. why should we care about pork quality used for human consumption?
Line 53: change to "affects"
Line 135: define for the readers what "FMT" stands for
Figures: authors should define the abbreviations: LPS, BBB etc. in text as they read, rather than at the end of the paragraph. It will make the figure easier to follow
Table 1: in what way do the prebiotics "improve growth performance". Please add detail
Line 484: define "HFD"
Author Response
Reviewer #2:
Comments and Suggestions for Authors
The authors aim to synthesise across relevant previous studies on pig gut microbiota, and fat and muscle development to discuss how microbiota may affect pork quality for human consumption. Consumers are becoming more concerned about the health of the animal whose meat they are consuming. Particularly, there are concerns revolving around antibiotic and steroid consumption in animals that have carry-over effects in human consumers. In the changing consumer market and consumer behaviour landscape, I believe this review may be of importance and serve as an encouraging stepping stone for studies that aim to ameliorate meat quality through both better animal husbandry and improved human health outcomes.
General comments:
I suggest the title is changed to: "The role of gut microbiota in the skeletal muscle development and fat deposition in pigs"
Response: Thank you for your comment. We have changed the title. (Revised manuscript: Line 4-5)
I would like to see the authors discuss what type of practises are currently used in the animal husbandry that are actually harming the health of pigs and their human consumers. E.g., historically, antibiotics have been used pre-emptively is animal feed to "proactively fight bacterial infections" (dangerous due to antimicrobial resistant (AMR) strain increase) and to increase body mass of the animal to produce more meat. This is an important part of the discussion and should be included in the review. I would also like to read more about how the different types of antibiotics affect the many positive effects of an intact and natural gut microbiota the farm pigs already have. I believe this part is vital considering the journal is called Antibiotics and the current part on antibiotics is too brief. Authors could also make note in the discussion of how AMR in pigs could serve as a dangerous source of zoonotic diseases
Response: Thank you for your review. We have enriched this issue. Indeed, the use of different types of antibiotics is associated with specific diseases, age or farm management. For example, penicillins were widely used for prophylaxis and treatment of septicaemia, respiratory and urinary tract infections in various animal species. Tetracyclines were commonly used to treat respiratory diseases caused by Actinobacillus pleuropneumonia and Pasteurella multocida. Colistin was most commonly used in gastrointestinal conditions of piglets and weaners, while tylosin in fatteners and sows [121]. However, large volumes of antibiotics used in food animals contribute to the emergence and spread of antimicrobial resistant (AMR) that lie in the food chain to propagate to consumer [122]. Thus, a detailed understanding of the current pattern of antibiotic use in livestock is essential to support optimal antibiotic use. This could decelerate the emergence of AMR in animal production. (Revised manuscript: Line 398-408)
The discussion should focus on how to improve current practises using the knowledge we have at the moment by providing example strategies for future
Response: Thank you for your suggestion. We should focus on how to improve current practises using the knowledge we have at the moment by providing example strategies for future. In the section 5, we mainly focus on the strategies of dietary composition or interventions in modifying gut microbiota.
Authors need to proofread that all bacterial species and strain names are in italics
Response: Thank you for your comment. We have proofread all bacterial species and strain names are in italics.
When discussing FMT, the authors should also indicate the potential cons of FMT, e.g., how it may disturb the co-adapted microbiota, how engraftment is usually imperfect, and how the FMT-introduced bacteria will oftentimes be outcompeted by local strains after a while
Response: Thank you for your suggestion. We have enriched this issue. Although its potential is exciting, there are obstacles to the use of FMT in practical applications. Factors restricting wider application of FMT include problems with donor selection, a lack of optimized methods for the preparation of the FMT, recipient genetics, lifestyle and microbiota composition. Indeed, both microbial diversity and the presence of specific species in the recipient microbiota have been suggested to affect FMT engraftment [91]. Thus, FMT efficacy is highly linked with donors, appropriate FMT protocol, recipient clinical status. (Revised manuscript: Line 292-298)
Considering how some breeds of pigs seem to have distinct gut microbiota that makes them more prone to obesity, it would be wise to discuss how artificial selection by humans for higher body mass and fat content in pigs has likely had a role in this. Role of domestication and breeding has not been discussed by the authors
Response: Thank you for your suggestion. Indeed, artificial selection by humans for higher body mass and fat content in pigs may play a role in this-some breeds of pigs seem to have distinct gut microbiota that makes them more prone to obesity. We focused on the effect host genetics on shaping the composition gut microbiota, providing a theoretical basis for pig breeding to change microbial composition.
Authors discuss "beneficial bacteria" when talking about pre- and probiotics. Are these beneficial for the pigs (will they be actually healthier?) or are they beneficial for us (we like a certain pork quality and these bacteria will give us the desired effect?)
Response: Thank you for your review. Certainly, pre- and probiotics are beneficial not only for the pigs but also for human. Pre- and probiotics can improve growth performance and immunity in pigs, which leads to increase pork quality for human consumption.
Figure 1 and 2 are in the wrong order. The first figure is indicated as Figure 2 and vice versa
Response: Thank you for your comment. We have revised this issue. (Revised manuscript: Line 160-162, 488-490)
Specific edits:
Line 18: change to "purpose is"
Response: Thank you for your review, we have revised the "purpose is" according to your comments. (Revised manuscript: Line 20)
Line 19: change to "the factors affecting gut microbiota composition will also be discussed"
Response: Thank you for your comment. We have made the suggested changes. (Revised manuscript: Line 21)
Line 21: change to "understanding between the relationship between gut microbiota and meat quality in pigs, and how manipulation of gut microbiota may contribute to increasing pork quality for human consumption"
Response: Thank you for your suggestion. we have made this revision. (Revised manuscript: Line 24-25)
Line 26: remove "as"
Response: Thank you for your comment. We have removed "as". (Revised manuscript: Line 29)
Line 30: I suggest the authors make a comparison between pork consumption and other meat consumption. I.e. why should we care about pork quality used for human consumption?
Response: Thank you for your suggestions. World meat production reached 337 million tonnes in 2020, up 45 percent, or 104 million tonnes compared with 2000. Although many species are raised for their meat, only three accounted for nearly 90 percent of the global production during the past two decades: chicken, pig and cattle. In 2020, world poultry meat production, global pig meat output, world bovine meat output reached 133.3 million tonnes, 109.2 million tonnes, 71.4 million tonnes respectively. According to the UN’s Food and Agriculture Organization, a powerful growth with an expected increase of 13% by 2030 will occur in pig husbandry, indicating that the pig industry plays a crucial role in the food supply chain and possesses a high economic impact. It is well known that pork quality determines the purchase decision of consumers and affects producers and retailers. However, meat quality defects still exist, which bring great economic losses to the fresh meat market. Thus, we should care about pork quality used for human consumption.
Line 53: change to "affects"
Response: Thank you for your comment. We have changed to "affects". (Revised manuscript: Line 56)
Line 135: define for the readers what "FMT" stands for
Response: Thank you for your suggestion. We have revised this issue. (Revised manuscript: Line 137)
Figures: authors should define the abbreviations: LPS, BBB etc. in text as they read, rather than at the end of the paragraph. It will make the figure easier to follow
Response: Thank you for your comment. We have defined the abbreviations: LPS, BBB etc. in text as they read. (Revised manuscript: Line 162-167)
Table 1: in what way do the prebiotics "improve growth performance". Please add detail
Response: Thank you for your comment. We have added the information in the table 1. (Revised manuscript: Line 457)
Line 484: define "HFD"
Response: Thank you for your comment. “HFD” defined on line 347.

Reviewer 3 Report
Manuscript ID 1752189 “Role of gut microbiota in pigs for skeletal muscle development and fat deposition: an overview”.
The article synthesizes much of the recent literature of animal that have demonstrated links between diet, and the gut microbiota.
I have a few minor suggestions and comments.
lines 157 and 159: Figure 2 ?
lines 468 and 470: Figure 1 ?
Lack sources of the citation search identified. It is crucial to divulge the databases that were searched in the article.
Check references for accuracy, completeness and consistency.
Author Response
Reviewer #3:
Comments and Suggestions for Authors
Manuscript ID 1752189 “Role of gut microbiota in pigs for skeletal muscle development and fat deposition: an overview”.
The article synthesizes much of the recent literature of animal that have demonstrated links between diet, and the gut microbiota.
I have a few minor suggestions and comments.
lines 157 and 159: Figure 2 ?
Response: Thank you for your comment. We have changed the order of Figure1 and Figure2. (Revised manuscript: Line160-162)
lines 468 and 470: Figure 1 ?
Response: Thank you for your comment. We have changed the order of Figure1 and Figure2. (Revised manuscript: Line 488-490)
Lack sources of the citation search identified. It is crucial to divulge the databases that were searched in the article.
Response: Thank you for your suggestion. Each reference has its own DOI number, so readers can search for them exactly.
Check references for accuracy, completeness and consistency.
Response: Thank you for your comment. We have carefully checked the references again.
